# The Effects of Use of Long-Term Second-Generation Antipsychotics on Liver and Kidney Function: A Prospective Study

**DOI:** 10.3390/diseases10030048

**Published:** 2022-07-22

**Authors:** Evangelia Papatriantafyllou, Dimitris Efthymiou, Maria Markopoulou, Efthymia-Maria Sakellariou, Emilia Vassilopoulou

**Affiliations:** 1Department of Nutritional Sciences and Dietetics, International Hellenic University, 57400 Thessaloniki, Greece; e.papatriantafyllou@hotmail.com; 2Department of Psychiatry, Division of Neurosciences, School of Medicine, Aristotle University of Thessaloniki, 54124 Thessaloniki, Greece; dimitrisefthy@gmail.com; 3Department of Forensic Psychiatry, Psychiatric Hospital of Thessaloniki, 56429 Thessaloniki, Greece; marmark33@yahoo.gr; 4Psychiatric Clinic, General University Hospital, 41110 Larissa, Greece; efthymiamaria@yahoo.com

**Keywords:** forensic psychiatry, antipsychotic drugs, liver, kidney

## Abstract

(1) Background: The second-generation antipsychotics (SGAPs) induce metabolic and inflammatory side effects, but documentation of their effects on the liver and kidneys is scarce. Aim: To study the three-year fluctuation of selected markers of renal and hepatic function in forensic psychiatric patients receiving SGAPs for more than five years. (2) Methods: Thirty-five forensic psychiatric patients (N = 35) were classified into two groups according to the type of SGAPs used for their treatment and the relevant risk of weight gain and metabolic complications. The three-year medication history, anthropometric data and biochemical data relevant to renal and hepatic function were retrieved from the individual medical files, specifically: serum levels of urea, uric acid, creatinine, alkaline phosphatase and amylase; the liver function enzymes, serum glutamic oxaloacetic transaminase (SGOT), serum glutamic pyruvic transaminase (SGPT) and gamma-glutamyl transpeptidase(γ-GT), and also the inflammatory index C-reactive protein (CRP). (3) Results: The patients receiving the SGAPs with a low risk for weight gain showed no significant fluctuation in the biochemical markers over the three-year period. The patients receiving the SGAPs with a high risk for weight gain showed significant differences between at least two measurements of uric acid (*p* = 0.015), SGOT (*p* = 0.018) and SGPT (*p* = 0.051). They showed significantly higher levels of creatinine in the third year compared to the second year (*p* = 0.029), and SGOT in the second year compared to the first (*p* = 0.038), and lower levels of SGPT in the third year compared to the second (*p* = 0.024). (4) Conclusion:In addition to consideration of possible metabolic and inflammatory complications, the choice of an antipsychotic drug for long-term treatment should also take into account the risk of hepatotoxicity and kidney damage.

## 1. Introduction

In forensic psychiatry, mentally disordered offenders are treated in a forensic psychiatric hospital, and their treatment often includes the administration of antipsychotic drugs (APs) [1]. APs are used to treat schizophrenia, bipolar disorder, monopolar depression, and are also commonly used in a number of other psychiatric disorders [2]. AP treatment for psychiatric disorders, as in forensic psychiatric patients, is often a long-term or even lifelong procedure. These patients are at high risk of AP-related adverse side effects, from both long-term and episodic treatment [3].

Various studies have shown that the most common serious disturbances caused by APs are weight gain (WG) [4,5], metabolic syndrome [6], diabetes mellitus (DM), cardiovascular disease (CVD) [7], acute kidney injury (AKI) [8,9] and chronic kidney disease (CKD) [10,11], and liver damage [12]. AKI survivors frequently develop CKD [13] and CVD is the main cause of mortality among CKD patients, both in adults and in children [14]. Recently, several studies for early detection of AKI and CKD have been reported in order to prevent progression of kidney disease and CVD events [15,16,17,18].

Severe metabolic complications in forensic psychiatric patients may be relevant to the multipharmacotherapy used for their treatment, in comparison to monotherapy, that is generally relevant to milder complications [19]. Medication can seriously affect liver function [20], as the liver is the major site of metabolism of drugs, including antidepressants (ADs) and APs [21]. More than 160 psychotropic drugs have been shown to produce hepatic side effects [22], and a recent study reported that psychotropic drugs were responsible for 7.6% of cases of drug-induced liver injury (DILI) in a group of 185 patients [23]. It is of note that both first-generation APs (FGAPs) and second-generation APs (SGAPs) are associated with a risk of hepatotoxicity [24].

Fluoxetine, paroxetine, sertraline, citalopram, and fluvoxamine are selective serotonin reuptake inhibitors (SSRIs) that have been associated with hepatotoxicity. The serotonin and norepinephrine reuptake inhibitors (SNRIs), venlafaxine, duloxetine, and trazodone, have also been strongly associated with hepatotoxic side effects [24]. Over-the-counter FGAPs, such as chlorpromazine and haloperidol, often cause serum elevation of the liver enzymes and severe liver damage. Of the SGAPs, clozapine is associated with hepatotoxicity [25]. An asymptomatic increase in serum transaminase levels is observed in up to 60% of patients treated with clozapine, with 15–30% showing a double or triple increase [26].

Moreover, observational studies have linked SGAPs to an increased risk of both AKI and CKD [8,9,10,11,12]. Reports have described an association of clozapine, olanzapine, and quetiapine with interstitial nephritis and AKI [27,28]. According to the study of Højlund and colleagues [12], all SGAPs, with the exception of aripiprazole, were associated with an increased risk of CKD. The risk was most pronounced for clozapine (aOR 1.81, 95% CI: 1.22 to 2.69) followed by olanzapine (aOR 1.41, 95% CI: 1.19 to 1.65) and quetiapine (aOR 1.28, 95% CI: 1.17 to 1.42) [12].

The aim of this study was to investigate the three-year fluctuation of selected markers of renal and hepatic function in forensic psychiatric patients receiving SGAP medication for more than five years, in relation to body weight, the serum level of C-reactive protein (CRP), and the type of AP.

## 2. Materials and Methods

### 2.1. Patients 

This prospective, three-year study included patients with psychotic disorders, all of whom were offenders, classified as not responsible for their criminal actions, and hospitalized in the Thessaloniki Psychiatric Hospital following a relevant judicial decision. The International Statistical Classification of Diseases (ICD-10) was used for the diagnosis of the mental disorders by qualified psychiatrists at the hospital. 

At the start of the study (T1), 49 patients were screened for eligibility to inclusion. Inclusion criteria included more than five years of SGAP use. After screening, 35 patients were eligible for participation, as 13 were receiving SGAPs for less than five years, while one had an incomplete medical record at T1. All patients were informed about the scope of the study and informed consent was obtained. The study was approved by the Scientific Committee of the hospital (code number: ∆3B/34800/21/07/2020) and the Research Ethics Committee of the Aristotle University of Thessaloniki (code number 4/26.01.2021), and complied with the International Code of Medical Ethics of the World Medical Association and the Helsinki Declaration. For the purposes of the study, patients were classified into two groups, based on the type of SGAP, as previously described in detail [1]: AP1, relevant to low to moderate risk of WG from the baseline weight and less risk of metabolic complications, namely aripiprazole, amisulpride, quetiapine, risperidone, and ziprasidone, and AP2, linked with a higher risk of WG from the baseline weight and a higher risk of metabolic complications, namely paliperidone, olanzapine, asenapine, and clozapine. Clinical information was collected from the hospital records of the patients, including the SGAPs used, changes in prescribed SGAPs over time, and medications used for the treatment of other chronic diseases, such as lipidemia, CVD, and type II DM. A flow chart of the study design is presented in Figure 1. 

SGAPs: second-generation antipsychotics. AP1: second-generation antipsychotics linked with a higher risk of weight gain (AP1). AP2: second-generation antipsychotics linked with a higher risk of weight gain (AP2). SGOT: serum glutamic oxaloacetic transaminase; SGPT: serum glutamic pyruvic transaminase; γ-GT: gamma-glutamyl transpeptidase; CRP: C-reactive protein.

### 2.2. Biochemical/Hematological Assessment 

Venous blood samples were obtained from all the participants, after overnight fasting, during the first quarter of the years in the three-year study period (i.e., T1: 2018, T2: 2019, T3: 2020), as part of their routine monitoring procedure. Serum and plasma aliquots were isolated and stored at −80 °C until further analysis.

The specific markers measured were the serum levels of urea, uric acid, creatinine, alkaline phosphatase and amylase, and the liver function enzymes, serum glutamic oxaloacetic transaminase (SGOT), serum glutamic pyruvic transaminase (SGPT), and gamma-glutamyl transpeptidase(γ-GT). Measurements were performed with an automatic analyzer (Toshiba TBA 120FR; Toshiba MedicalSystems Co., Ltd., Tokyo, Japan) under standard conditions, on the hospital premises.

### 2.3. Anthropometric Measurements

On the morning of blood collection, all participants underwent anthropometric measurements, after fasting for at least 8 h, by one trained investigator. A commercial stadiometer (Leicester Height Measure, Invicta Plastics Ltd., Oadby, UK) was used to measure height to the nearest 0.1 cm; height was measured with the participants barefoot, their shoulders in a relaxed position, their arms hanging freely, and their heads in the Frankfort horizontal plane. Weight was measured with the participants barefoot and in light clothing to the nearest 0.1 kg using a TANITA RD-545 (“RD-545-Connected smart scale|Tanita Official Store”, n.d.). Body mass index (BMI) was calculated from the current weight and height, as weight (kg) by height squared (m^2^). The waist circumference (WC) was measured with a SECA flexible, inextensible measuring tape, with an accuracy of 1 mm, on a horizontal plane, after exhalation, at a point equidistant from the lowest floating rib and the upper border of the iliac crest.

### 2.4. Statistical Analysis

The statistical analysis was performed with the SPSS23 package. The categorical values were expressed as mean ± standard deviation (SD), while the non-categorical values as median (interquartile range (IQR)). The possible differences between biochemical characteristics at two time periods for 2dependent samples was determined with the t-test or with the Wilcoxon test, depending on the regularity of the values. Significant differences between the groups of patients, AP1 and AP2, based on the SGAP drug they were receiving, were explored by repeated measures ANOVA. When the differences in observations violated the hypothesis of regularity, the difference tests were performed with the Friedman method.

Multiple regression analysis was performed to explore the effect of weight, BMI, and CRP on changes in the renal and hepatic biochemical markers.

## 3. Results

Of the 49 forensic psychiatric patients initially assessed in the study, 35 fulfilled the criteria for inclusion; among the excluded, 13 received SGAPs for less than 5 years, while 31 (91%) were males, and 17 were receiving AP1 and 18 AP2 medication. Their demographic and clinical characteristics are presented in Table 1. The two groups had a similar age range (AP1: 52.35 ± 12.68 years; AP2: 48.56 ± 12.65 years, *p* = 0.38). All patients received the same SGAP over the study period (i.e., no in-person changes in SGAP received). The patients receiving AP2 medication more frequently used anti-diabetic and anti-lipidemic medication, and medication for CVD (i.e., heart rate lowering agents, such as ivabradine, beta blockers, such as propranolol, and calcium channel blockers, such as amlodipine), than the patients receiving AP1.

The AP1 group presented no changes over the study period in the biochemical markers measured (Table 2).

Patients in the AP2 group presented a significant difference in serum level of uric acid (*p* = 0.01), SGOT (*p* = 0.02), and SGPT (*p* = 0.05) in at least one measurement over the study period (Table 3). Specifically, AP2 patients showed significantly higher mean uric acid level in the T3 than in the T2 measurement (*p* = 0.01), and significantly higher creatinine level in the T3 than in the T1 measurement (*p* = 0.03). SGOT levels in the AP2 patients were significantly higher in T2 than in the T1 measurement (*p* = 0.04), and SGPT levels were significantly lower in the T3 than in the T2 measurement (*p* = 0.02).

Multiple regression analysis was conducted to determine whether the differences in uric acid, creatinine, SGOT, and SGPT in the AP2 participants were affected by weight, BMI, and CRP (Table 4). Due to the small number of individuals in group AP2 (18), all 3 measurements over time for each individual were examined for every determinant.

The first regression model demonstrated the significance for prediction of increase in uric acid level in AT2 patients of weight (t (53) = 2.564, *p* = 0.01) and BMI (t (53) =−2.012, *p* = 0.05), explaining 14.1% of its total variability (F (3.50) = 2.741, *p* = 0.05). Specifically, uric acid level showed a positive correlation with weight (B = 0.061) and a negative correlation with BMI (B = −0.134). Therefore, AP2 patients are expected to have a higher uric acid level as their weight increases while BMI and CRP levels remain stable. In contrast, AP2 patients who have a higher BMI, but constant weight and CRP, are expected to show a decrease in uric acid level.

The third model demonstrated significance in predicting SGOT levels (F(3.50) = 3.599, *p*= 0.02) explaining 17.8% of the total variability. Specifically, a positive correlation was shown between BMI and SGOT levels (B = 0.816), and BMI showed significance for predicting SGOT levels (t(53) = 2.141, *p*= 0.03). This means that for AP2 patients with the same weight and CRP level, higher SGOT levels can be expected as BMI increases.

## 4. Discussion

Schizophrenia and other severe psychiatric disorders show reciprocal association with metabolic disturbances [29,30]; obesity, DM, and metabolic syndrome are common comorbidities in patients with schizophrenia, especially those receiving AP treatment [7,31]. APs are strongly associated with the core components of metabolic syndrome, namely WG, glucose intolerance, and dyslipidemia [7,32]. More advanced adverse metabolic events are related to the induction of liver injury [33]. In this study, we showed that the SGAPs known to induce more significant metabolic complications [1], referred to here as the AP2 group, are related to the severity of impairment of renal and hepatic function.

Recently, Druschky and his colleagues (2020) [34], assessing in a cross-sectional study the liver injury provoked by the use of APs, concluded that olanzapine, perazine, and clozapine were responsible for the most severe drug-induced liver damage. Similarly, we observed a positive association of elevated hepatic biochemical markers with the ongoing use of AP2 drugs, including paliperidone, olanzapine, asenapine, and clozapine. Conversely, the AP1 drugs, namely aripiprazole, amisulpride, quetiapine, risperidone, and ziprasidone, did not induce significant alterations in the liver function indices.

Generally, the antipsychotic drugs, together with neurologic, are referred to as the second most hepatotoxic drugs, after anti-infectious drugs [35]. To explain this sideeffect, several factors have been identified as important, because they affect the blood concentration of the drug, and therefore, it becomes toxic for the liver. Particularly, the pharmacokinetics of the drug and the first pass metabolism; the protein binding; the metabolic routes of the drug, and the body fluid status. In a study by Cho and colleagues [36], WG, rather than weight variability, was reported to induce liver dysfunction and non-alcoholic fatty liver disease (NAFLD) [36]. Of the SGAPs, the AP2, in contrast to the AP1 drugs, provoke rapid increase of body weight which is generally considered as the main inducer of [37]. The AP2 drugs induce metabolic complications, including hyper-triglyceridemia, hypercholesterolemia, and an increase in serum low density lipoprotein (LDL), even in the absence of WG [38,39]. There are also indications that subjects with dyslipidemia have a higher likelihood of developing liver disease than subjects without dyslipidemia [40].

Kidney dysfunction, also, is associated with dyslipidemia, specifically elevated levels of triglycerides, low levels of high-density cholesterol (HDL) and increased levels of small dense LDL, along with an increased CVD risk [41]. Long-term SGAP use has been linked with increased risk of chronic kidney dysfunction, regardless the age of the subjects using these drugs [12]. Additionally, use of SGAPs, and especially olanzapine, quetiapine, zotepine or risperidone, has been proposed as an exacerbator for the risk of CKD, especially in the presence of other comorbidities such as dyslipidemia, obesity, and DM [11]. Specifically, clozapine and quetiapine have been linked with interstitial nephritis and AKI [27,28]. In line, the forensic psychiatric patients in our case series were overweight, and those receiving AP2 were found to be at higher risk for major change in kidney markers.

It is of note that the increase in the level of uric acid in the AP2 patients was correlated with an increase in the CRP level, and with WG. Metabolic syndrome components, including WG, have been previously associated with high levels of CRP and uric acid [42], while CRP is highly predictive of subsequent risk of cardiovascular events and DM in apparently healthy men and women [43]. The correlation in our study of uric acid and CRP with administration of AP2 drugs supports our previous suggestion that this group of SGAPs is responsible for extensive metabolic and inflammatory complications [1,42,44].

Maintenance treatment for chronic illnesses, such as psychiatric diseases, is often long-term or lifelong, making drug selection a difficult process, due to the risk of extensive side effects in the function of various organs. We present here evidence that the use of SGAPs, and specifically of AP2s, can have adverse effects on liver and kidney function. Multiple regression analysis was performed to explore the contribution of weight, BMI, and level of CRP.

Our prospective study on the long-term side effects of SGPAs in a group of mentally disordered offenders offers significant distinguished information relevant to the specific type of AP, as these were categorized by the risk of weight gain and metabolic complications they provoke. As the patients were hospitalized, they were all following the same diet plan, and as such, the effect of diet on the specific markers was limited. Nevertheless, the study has some limitations. Due to the small study sample size, this analysis was conducted for the whole of the study period and not for the individual time points. In order to draw more robust conclusions on the hepatotoxic and nephrotoxic effect of the AP2 group of SGAPs, further investigation is needed with larger samples. Special consideration should be given to populations such as forensic psychiatric patients, where other risk factors may amplify complications, for instance, multi-drug treatment, drug abuse, obesity, metabolic syndrome, and DM. Moreover, although the specific doses of SGAPs were recorded during our data collection, as well as the doses of the other drugs, these were not taken into account in the current investigation. Similarly, the specific effect on the liver or kidney function of the individual SGAP used in the study could not be established due to the small sample size. Another limitation of this study which evolved due to the small number of participants is that the regression analysis was run on a sample of 54 values which were not independent, and this is a limitation for drawing safe conclusions. The results of the study, however, indicate possible adverse effects on the kidney and liver of long-term administration of the AP2 group SGAPs, a finding which warrants further study in patients using this group of drugs.

## 5. Conclusions

Our study shows that the long-term use of SGAPs linked with the induction of weight gain, are more capable to provoke liver and kidney dysfunction, both relevant to the metabolic and inflammatory complications induced by them. Further investigation in higher study samples are needed to confirm and expand our results for proper treatment and prevention of these complications.

## Figures and Tables

**Figure 1 diseases-10-00048-f001:**
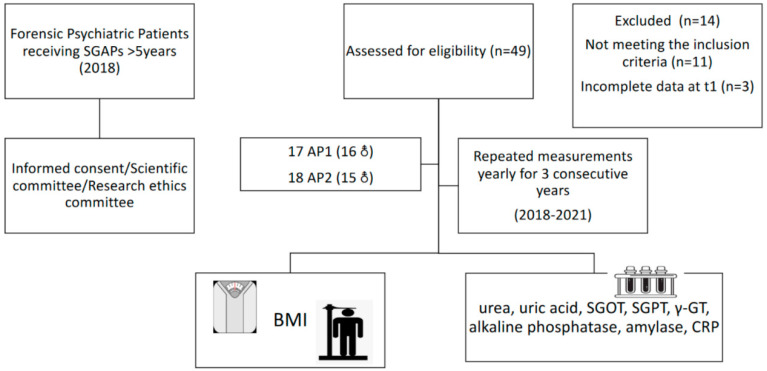
Flow chart of the prospective study.

**Table 1 diseases-10-00048-t001:** Demographic and clinical characteristics of the forensic psychiatric patients at the start of the study (T1) (N = 35), according to the type of antipsychotic (AP) medication.

	AP1	AP2
N	17	18
Males	16 (94%)	15 (83.33%)
Antipsychotics	17	18
Anticoagulants	5	1
Antihypertensives	3	4
Antidiabetic	2	3
Antilipidemic	2	5
Medication for CVD	7	7

AP1: Antipsychotic medication associated with low risk of weight gain; AP2: Antipsychotic medication associated with higher risk of weight gain; CVD: cardiovascular disease.

**Table 2 diseases-10-00048-t002:** Changes (difference tests) in the biochemical measurements of forensic psychiatric patients taking second-generation antipsychotics with a low risk of weight gain (AP1) over the three-year study period (time points T1, T2, T3) (N = 17).

Biochemical Measurement	T1	T2	T3	T1-T2 *p*-Value	T2-T3 *p*-Value	T1-T3 *p*-Value	Overall Difference during the 3-Year Period *p*-Value
Urea (mg/dL)	27.12 (11.77)	27.35 (9.58)	27.29 (8.34)	0.89	0.97 **	0.94 **	0.79
Uric acid (mg/dL)	4.19 (1.52)	4.20 (1.35)	4.30 (1.70)	0.067 **	0.97	0.12 **	0.06
Serum creatinine (mg/dL)	0.83 (0.24)	0.81 (0.15)	0.80 (0.24)	0.88 **	0.28 **	0.400 **	0.53 ***
SGOT (U/L)	17.00 (7.00)	18.00 (9.00)	17.00 (10.00)	0.79 **	0.48	0.85 **	0.34
SGPT (U/L)	20.00 (11.00)	18.00 (17.00)	21.00 (19.00)	0.22	0.26	0.14 **	0.39
γ-GT (U/L)	26.00 (13.00)	20.00 (13.00)	24.00 (18.00)	0.63 **	0.12	0.56 **	0.48
Akaline phosphatase (U/L)	79.47 (31.35)	79.29 (31.47)	80.41 (36.93)	0.93 **	0.71 **	0.81 **	0.86 ***
Amylase (U/L)	58.29 (19.66)	57.41 (21.30)	62.12 (14.32)	0.57	0.08	0.27 **	0.19

SGOT: serum glutamic oxaloacetic transaminase; SGPT: serum glutamic pyruvic transaminase; γ-GT: gamma-glutamyl transpeptidase. All other values are expressed as median (intra-quaternary range, IQR). ** *t* test was applied for 2 dependent samples; the non-parametric Mann–Whitney test was used for all other samples. *** tests were performed with the parametric test repeated measures ANOVA. All other tests were performed using the non-parametric Friedman test.

**Table 3 diseases-10-00048-t003:** Changes (difference tests) in the biochemical measurements of forensic psychiatric patients taking second-generation antipsychotics with a higher risk of weight gain (AP2) over the three-year study period (time points T1, T2, T3) (N = 18).

Biochemical Measurement	T1	T2	T3	T1-T2 (*p*-Value)	T2-T3 (*p*-Value)	T1-T3 (*p*-Value)	Overall Difference during the 3-Year Period (*p*-Value)
Urea (mg/dL)	24.03 (8.46)	23.31 (8.52)	23.77 (10.71)	0.39 **	0.81 **	0.86 **	0.94
Uric acid (mg/dL)	4.21 (1.40)	4.00 (2.18)	4.50 (1.55)	0.60	0.01 **	0.23 **	0.01
Serum creatinine (mg/dL)	0.81 (0.12)	0.84 (0.30)	0.87 (0.31)	0.13	0.41 **	0.03 **	0.16
SGOT (U/L)	15.00 (4.00)	17.00 (2.00)	15.00 (4.00)	0.04	0.01	0.82	0.02
SGPT (U/L)	17.00 (12.00)	18.50 (16.00)	14.00 (13.00)	0.24 **	0.02 **	0.06	0.05
Γ-gt (U/L)	24.50 (18.00)	24.50 (17.00)	24.00 (9.00)	0.72	0.19	0.42 **	0.54
Alkaline phosphatase (U/L)	73.28 (18.10)	72.22 (21.89)	74.56 (22.56)	0.63 **	0.33	0.41	0.09
Amylase (U/L)	49.94 (21.84)	51.94 (19.33)	53.89 (22.13)	0.67	0.78	0.86	0.86

SGOT: serum glutamic oxaloacetic transaminase; SGPT: serum glutamic pyruvic transaminase; γ-GT: gamma-glutamyl transpeptidase. All other values are expressed as median (intra-quaternary range, IQR). ** *t* test was applied for 2 dependent samples; the non-parametric Mann–Whitney test was used for all other samples. Non-parametric Friedman test was used for all other tests.

**Table 4 diseases-10-00048-t004:** Regression analysis of specific factors for the forensic psychiatric patients taking second-generation antipsychotics with a higher risk of weight gain (AP2).

Dependent Variable	Independent Variables	Β	S.E.	β	T	*p*
Uric acid ^(1)^	Constant	3.013	1.278		2.358	0.02
	Weight	0.061	0.024	0.546	2.564	0.01
	BMI	−0.134	0.067	−0.432	−2.012	0.05
	CRP	−0.048	0.051	−0.124	−0.932	0.36
Serum creatinine ^(2)^	Constant	0.482	0.174		2.761	0.01
	Weight	0.001	0.003	0.066	0.303	0.76
	BMI	0.012	0.009	0.280	1.281	0.21
	CRP	−0.003	0.007	−0.050	−0.367	0.71
SGOT ^(3)^	Constant	4.205	7.290		0.577	0.57
	Weight	−0.124	0.135	−0.190	−0.914	0.36
	BMI	0.816	0.381	0.450	2.141	0.03
	CRP	0.496	0.292	0.222	1.698	0.09
SGPT ^(4)^	Constant	−1.128	8.832		−0.128	0.89
	Weight	0.011	0.164	0.014	0.065	0.95
	BMI	0.733	0.462	0.342	1.587	0.12
	CRP	−0.366	0.354	−0.139	−1.033	0.31

BMI: body mass index; CRP; C-reactive protein; SGOT: serum glutamic oxaloacetic transaminase; SGPT: serum glutamic pyruvic transaminase; ^(1)^ F(3,50) = 2.741, *p* =0.05, R^2^ = 0.141; ^(2)^ F(3,50) = 2.059, *p* =0.12, R^2^ = 0.110; ^(3)^ F(3,50) = 3.599, *p* =0.02, R^2^ = 0.178; ^(4)^ F(3,50) = 2.524, *p* =0.06, R^2^ = 0.132.

## Data Availability

Data are available upon request to the correspondent authors.

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
