# Peer review of "The Effects of Use of Long-Term Second-Generation Antipsychotics on Liver and Kidney Function: A Prospective Study"

_diseases, 2022, doi:10.3390/diseases10030048_

Round 1

Reviewer 1 Report

Introduction.

1. Please avoid paragraphs in this section as it is not a narrative review. In other words, you must introduce the study.

2. The sentence "The specific markers examined were the
serum levels of urea, uric acid, creatinine, alkaline phosphatase and amylase, and the liver function enzymes serum glutamic oxaloacetic transaminase (SGOT), serum glutamic pyruvic transaminase (SGPT) and gamma-glutamyl transpeptidase -GT)" can be moved in the methods.

Methods.

1. Was the informed consent obtained?

2. Since it is a prospective observational investigation, a flow chart of the study must be provided.

3. Please offer details on inclusions/exclusion criteria used.

4. What about doses, and other variable? Please discuss in the limitations

5. What about different drugs? Please discuss in the limitations.

6. It seems to be a retrospective analysis.

Discussion

1. The discussion needs to be better focused on your findings and on what is reported in the literature

Moreover:

1. Please specify the type of the study in the title.

2. Add a Conclusions section

Author Response

Reviewer1

Introduction.

  1. Please avoid paragraphs in this section as it is not a narrative review. In other words, you must introduce the study.

Corrected. Thank you for the recommendation.

  1. The sentence "The specific markers examined were the serum levels of urea, uric acid, creatinine, alkaline phosphatase and amylase, and the liver function enzymes serum glutamic oxaloacetic transaminase (SGOT), serum glutamic pyruvic transaminase (SGPT) and gamma-glutamyl transpeptidase (γ-GT)" can be moved in the methods. Thank you, corrected.

Methods.

  1. Was the informed consent obtained?

An informed consent was obtained and the relevant information is included in the methods section (lines 35-37)

  1. Since it is a prospective observational investigation, a flow chart of the study must be provided.

A flow chart is inserted following your recommendation

  1. Please offer details on inclusions/exclusion criteria used. We made proper amendments (lines 95-103)
  2. What about doses, and other variable? Please discuss in the limitations We have inserted relevant information
  3. What about different drugs? Please discuss in the limitations. We have inserted relevant information
  4. It seems to be a retrospective analysis

The study was prospective in design. Nevertheless, the specific outcome was a secondary outcome. The first outcome is presented in detail elsewhere

(Vassilopoulou, E.; Efthymiou, D.; Papatriantafyllou, E.; Markopoulou, M.; Sakellariou, E.-M.; Popescu, A. C. Long Term Metabolic and Inflammatory Effects of Second-Generation Antipsychotics: A Study in Mentally Disordered Offenders. J.

Pers. Med. 2021, 11 (11), 1189. https://doi.org/10.3390/jpm11111189)

Discussion

  1. The discussion needs to be better focused on your findings and on what is reported in the literature

Moreover:

  1. Please specify the type of the study in the title. Thank you
  2. Add a Conclusions section Thank you

Reviewer 2 Report

This manuscript is interesting; however, there are some points to be revised.

1) "Creatinine" in Tables 2, 3, and 4 means serum creatinine? if so, the authors clearly describe it.

2) Introduction (Page 2, Lines 1-3): "APs are weight gain (WG) [4,5], metabolic syndrome [6], diabetes mellitus (DM), cardiovascular disease (CVD) [7] acute kidney injury (AKI) [8,9] and chronic kidney disease (CKD) [10,11], and liver damage [12]."

After these descriptions, the authors should add "AKI survivors frequently develop CKD (Int J Mol Sci 2021 Oct 14;22(20):11093) and CVD is the main cause of mortality among CKD patients, both in adults and in children (Biomedicines. 2022 Jun 13;10(6):1396). Recently, several studies for early detection of AKI and CKD have been reported in order to prevent progression of kidney disease and CVD events (Lancet. 2005 Apr 2-8;365(9466):1231-8;Kidney Int. 2006 Jul;70(1):199-203; J Pharmacol Exp Ther. 2012 Jun;341(3):656-62; Clin J Am Soc Nephrol. 2020 Mar 6;15(3):349-358)."

Author Response

Reviewer2

This manuscript is interesting; however, there are some points to be revised.

  • "Creatinine" in Tables 2, 3, and 4 means serum creatinine? if so, the authors clearly describe it.

Proper corrections are applied. Thank you 

  • Introduction (Page 2, Lines 1-3): "APs are weight gain (WG) [4,5], metabolic syndrome [6], diabetes mellitus (DM), cardiovascular disease (CVD) [7] acute kidney injury (AKI) [8,9] and chronic kidney disease (CKD) [10,11], and liver damage [12]."

After these descriptions, the authors should add "AKI survivors frequently develop CKD (Int J Mol Sci 2021 Oct 14;22(20):11093) and CVD is the main cause of mortality among CKD patients, both in adults and in children (Biomedicines. 2022 Jun 13;10(6):1396). Recently, several studies for early detection of AKI and CKD have been reported in order to prevent progression of kidney disease and CVD events (Lancet. 2005 Apr 2-8;365(9466):12318;Kidney Int. 2006 Jul;70(1):199-203; J Pharmacol Exp Ther. 2012 Jun;341(3):656-62; Clin J Am Soc Nephrol. 2020 Mar 6;15(3):349-358).

Thank you for the useful contribution to our manuscript’s content.

Round 2

Reviewer 1 Report

I endorse it